# The lost world of Cuatro Ciénegas Basin, a relictual bacterial niche in a desert oasis

Valeria Souza[1†]*, Alejandra Moreno-Letelier[2†], Michael Travisano[3], Luis David Alcaraz[4‡], Gabriela Olmedo[5], Luis Enrique Eguiarte[1]*

[1]Departamento de Ecología Evolutiva, Instituto de Ecología, Universidad Nacional Autónoma de México, Coyoacán, Mexico; [2]Jardín Botánico, Instituto de Biología Universidad Nacional Autónoma de México, Coyoacán, Mexico; [3]Department of Ecology, Evolution and Behavior, University of Minnesota, Saint Paul, United States; [4]Laboratorio Nacional de la Ciencias de la Sostenibilidad, Instituto de Ecología, Universidad Nacional Autónoma de México, Coyoacán, Mexico; [5]Laboratorio de Biología Molecular y Ecología Microbiana, Departamento de Ingeniería Genética, Unidad Irapuato Centro de Investigación y Estudios Avanzados, Guanajuato, Mexico

**\*For correspondence:**
souza@unam.mx (VS);
fruns@unam.mx (LEE)

[†]These authors contributed equally to this work

**Present address:**
[‡]Departamento de Biología Celular, Facultad de Ciencias, Universidad Nacional Autónoma de México, Mexico City, Mexico

**Competing interests:** The authors declare that no competing interests exist.

**Abstract** Barriers to microbial migrations can lead adaptive radiations and increased endemism. We propose that extreme unbalanced nutrient stoichiometry of essential nutrients can be a barrier to microbial immigration over geological timescales. At the oasis in the Cuatro Ciénegas Basin in Mexico, nutrient stoichiometric proportions are skewed given the low phosphorus availability in the ecosystem. We show that this endangered oasis can be a model for a *lost world*. The ancient niche of extreme unbalanced nutrient stoichiometry favoured survival of ancestral microorganisms. This extreme nutrient imbalance persisted due to environmental stability and low extinction rates, generating a diverse and unique bacterial community. Several endemic clades of *Bacillus* invaded the Cuatro Cienegas region in two geological times, the late Precambrian and the Jurassic. Other lineages of *Bacillus*, *Clostridium* and *Bacteroidetes* migrated into the basin in isolated events. Cuatro Ciénegas Basin conservation is vital to the understanding of early evolutionary and ecological processes.
DOI: https://doi.org/10.7554/eLife.38278.001

## Introduction

A 'lost world' is both a poetic metaphor and a scientific idea; in both cases, the term pertains to the conservation or re-creation of the deep past in a particular place. Scientists have looked for analogs of such worlds in environments possessing living microbial mats and stromatolites, since these organized forms of life were dominant for billions of years during the Proterozoic (*Nutman et al., 2017*). Nevertheless, in most cases these communities represent more a physical metaphor of the past than an actual lost world, since they contain mostly contemporary microbial lineages (*White et al., 2015*; *Wong et al., 2015*). The exception seems to be the abundant and morphologically diverse stromatolites and microbial mats found in the endangered oasis of Cuatro Ciénegas Basin (CCB) in Northern Mexico. In this extremely diverse wetland (*Minckley, 1969*), the recycling of the deep aquifer by magmatic heat replicates many conditions of ancient oceans (*Wolaver et al., 2013*), including its extremely unbalanced nutrient stoichiometry (*Elser et al., 2006*) and sulphur and magnesium minerals that replicate marine osmolarity (*Wolaver et al., 2013*; *De Anda et al., 2017*; *Rebollar et al., 2012*), despite being low in NaCl. Moreover, isotopic analysis suggests that the deep aquifer maintained the ancestral marine conditions in the wetland by dissolving the existing minerals from its

**eLife digest** Water is a rare sight in a barren land, but there are many more reasons that make the Cuatro Cienegas Basin, an oasis in the North Mexican desert, a puzzling environment. With little phosphorous and nutrients but plenty of sulphur and magnesium, the conditions in the turquoise blue lagoons of the Basin mimic the ones found in the ancient seas of the end of the Precambrian. In fact, Cuatro Cienegas is one of the rare sites where we can still find live stromatolites, a bacterial form of life that once dominated the oceans. Many bacteria of marine origin exist alongside these living fossils, prompting scientists to wonder if the Basin could be a true lost world, a safe haven where ancient microorganisms found refuge and have kept evolving until this day. But to confirm whether this is the case would require scientists to hunt for clues within the genetic information of local bacteria.

Souza, Moreno-Letelier et al. came across these hints after sampling for bacteria in a small (about 1km$^2$) lagoon named Churince, and analysing the DNA collected. The results yielded an astonishing amount of biodiversity: 5,167 species representing at least two-third of all known major groups of bacteria were identified, nearly as much as what was found in over 2,000 kilometres in the Pearl River in China. This is unusual, as most other extreme environments with little nutrients have low levels of diversity.

Closer investigation into the genomes of 2,500 species of *Bacillus* bacteria revealed that the sample increased by nearly 21% the number of previously known species in the group. Most of these bacteria were only found in the Basin. These native or 'endemic' species have evolved from ancestors that came to the area in two waves. The oldest colonization event happened 680 million years ago, as the first animal forms just started to emerge. The most recent one took place while dinosaurs roamed the Earth about 160 million years ago, when geological events opened again the Basin to the ancient Pacific Ocean.

Previous experiments have shown that different species of bacteria in the Churince have evolved to form a close-knit community which ferociously competes with microbes from the outside world. Paired with the extreme conditions found in the lagoon, this may have prevented other microorganisms from proliferating in the environment and replacing the ancient lineages.

The days of this lost world may now be numbered. Drained by local farming, the wetlands of the Basin have shrunk by 90% over the past five decades. The Churince lagoon, the most diverse and fragile site where the samples were collected, is now completely dry. Human activities also disrupt the delicate and unique balance of nutrients in the oasis. But all may not be lost – yet. Local high school students have become involved in the research effort to describe and protect these unique microbial communities, and to change agricultural traditions in the area. Closing the canals that export spring water out of the Basin could give the site a chance to recover, and the microbes that are now seeking refuge in underground waters could re-emerge. Maybe there will still be time to celebrate, rather than mourn, the unique life forms of the Cuatro Cienegas Basin.

DOI: https://doi.org/10.7554/eLife.38278.002

sediments (*Wolaver et al., 2013*). These specific conditions along with an extreme unbalanced nutrient stoichiometry between nitrogen (N) and phosphorus (P), created a unique niche that has persisted (*Elser et al., 2006*). Ecological analyses have revealed that a 16:1 nitrogen to phosphorus (the Redfield ratio) is common to most life on Earth (*Elser et al., 2006*). However, at the oasis of CCB, such proportions are skewed given the low phosphorus in the ecosystem. We believe that these niche variables can explain the survival in this oasis of ancient marine bacteria and hydrothermal vent-associated sulphur microbes (*Souza et al., 2006*). Our hypothesis is that such marine microbes have stayed there for hundreds of millions of years (*Torsvik, 2003*), since this site was on the coasts of Laurentia for a very long time. This changed 35 mya, with the uplifts of the Sierras that isolated CCB from the Western Seaway (*Souza et al., 2006*) and the aridification of the Chihuahuan desert in the last 7 million years. In this 'lost world', even the most dynamic part of the community, the viruses, have maintained a marine signature, as viral metagenomics revealed substantial divergence of viruses from continental waters and a strong similarity with those of marine habitats (*Desnues et al., 2008*; *Taboada et al., 2018*).

There is a very high ratio of nitrogen to phosphorus (167:1) in the sediment of the Churince hydrological system, where most of the *Bacillus* of this study were sampled (*Lee et al., 2017*). We see an imprint of this evolutionary history in the extreme imbalance at the bacterial intracellular level in many lineages (the most extreme being nitrogen to phosphorus ratio of 965:1 in a strain of CCB *Bacillus cereus* group) (*Valdivia-Anistro et al., 2015*) and in the capability of some CCB *Bacillus* species to synthesize membrane sulfolipids, in what appears to be an ancestral adaptation to limited phosphorus availability acquired a long time ago from cyanobacteria by horizontal gene transfer (*Alcaraz et al., 2008*). Extreme unbalanced nutrient stoichiometry, as well as rich sulphur conditions, are niche characteristics of the Precambrian ocean, that ended abruptly at the onset of the Phanerozoic Eon 542 mya with the weathering of continental apatite as consequence of several glaciation events (*Planavsky et al., 2010*). Moreover, using conservative time estimates based on geological events, molecular clock studies have suggested that some strains of culturable cyanobacteria (*Domínguez-Escobar et al., 2011*) as well as of *Bacillus* (*Moreno-Letelier et al., 2012*) from CCB diverged also from their close relatives in the late Precambrian.

Hence, we propose that CCB is a microbial lost world, not just as a poetic metaphor, but as a real geographical site: a nutrient-unbalanced multidimensional niche isolated from the human environment. What would make a lost world more than a metaphor? Typically, there are three models of diversification (*Figure 1*). The null model for microbes is 'everything is everywhere and the environment selects'. We can observe this to apply for cosmopolitan bacteria, such as *Escherichia coli* or *Bacillus subtilis*, since these microbes have an enormous population size and considerable migration rates (*Figure 1a*). The standard model of biogeography is isolation by distance, a pattern similar to the one observed for most macro-organisms. This isolation has been observed for the thermophile Crenarchaeota *Sulfolobus islandicus* (*Reno et al., 2009*) (*Figure 1b*). The third is the island model of localized adaptation and rare migration (*Figure 1c*). This pattern also occurs in microbes, as is the case for those in the lakes in the Pyrenees, explained by the island-like nature of each lake (*Casamayor, 2017*). Here, we suggest that in order to explain CCB singularity, we need a 4th model, the lost world model (*Figure 1d*). A lost world would be both a physical space and a refugia, where communities survive in relictual conditions. One signature for a lost world would be the presence of very deep phylogenetic branches, given that the time since isolation would be expected to be very long and extinctions rare. The other would be extreme niche conservatism, in this case by strong environmental filtering given the extremely unbalanced nutrient stoichiometry.

## Results

In order to explore which of these four models fits our study site and its microbiota, we will first describe our site and the total microbial diversity we found in it. Churince is a closed hydrological system (and the most endangered site within CCB) and depends on recharge by the deep aquifer contained in the Sierra San Marcos (*Wolaver et al., 2013*). The system used to consist of a spring, an intermediate lagoon and a large desiccation lagoon connected by a river (*Minckley, 1969*). By the time we surveyed its microbial biodiversity, the desiccation lagoon had already disappeared. We obtained samples of environmental DNA from water, sediment and soil in different sampling points from the spring to the end of the intermediate lagoon, as well as from soil associated to different vegetation. We observed a vast microbial diversity within roughly a square kilometer, through PCR amplification and sequencing of 16S rRNA genes from environmental DNA (*Figure 2*). This diversity can be expressed in operational taxonomic units (OTUs), used to classify groups of related individuals that have 16S rRNA gene sequences exhibiting at least 97% identity. The Churince's total Bacteria and Archaea richness is represented by a total of 5,167 OTUs assigned to samples from the water column, aquatic sediments, and soil. These assigned OTUs represented 60 different known phyla, three of which were Archaea. Even though each site seems to have a unique taxonomic 'fingerprint' (*Figure 3*), despite their spatial closeness, we also observed general patterns that aquatic sites share, such as predominance of Proteobacteria, Actinobacteria, and Bacteroidetes. Sediments and soils are much more diverse, and have important phyla in larger proportions than water sites, such as Firmicutes (the phyla that encompasses *Bacillus* and *Clostridium)* and the primary producer Cyanobacteria. In sediments, Cyanobacteria are part of the microbial mats along with Chlorobi and Spirochaetes, while in the soil, Cyanobacteria are part of the microbial crusts where Acidobacteria also play an important role in nutrient cycling along with Nitrospira (*Figure 3*). It is important to

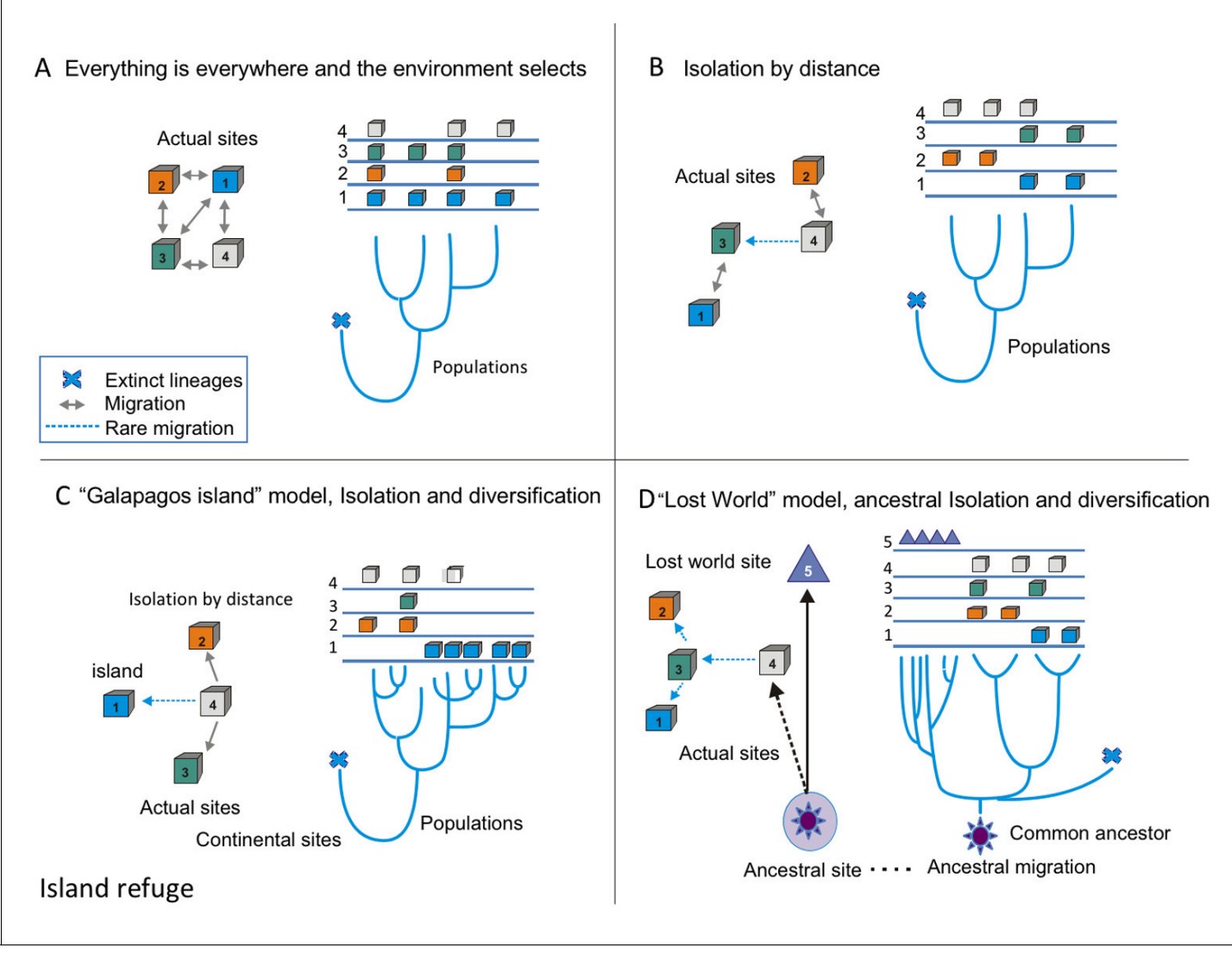

**Figure 1.** Conceptual frame-work of species diversification. (A) 'Everything is everywhere, and the environment selects' model implies that free migration is only restricted by environmental filtering. Hence, sites 1–4 have the same probability of migration, but some sites, such as one are better than others. In this case, the phylogenetic tree does not reflect the geographic structure; this is common in many cosmopolitan lineages of bacteria and fungi. (B) Model of isolation by distance, this is what occurs in most plants and animal phyla, sites that are closer (1 and 3 or 2 and 4) are more likely to present migration events than sites that far apart, some rare events of migration are allowed (as between 4 and 3). In this case, the branches of the tree reflect the geographic structure. (C) Island model implies that rare events of migration from the source, (4) to an island, such as Galapagos (1). The phylogentic tree reflects adaptive radiation due to isolation. (D) 'Lost world model' of ancestral isolation and diversification implies that lineages that were extinct in other places have remained as relictual niches persist in a new site (5). In this case, the ancestral lineages have very long branches that show their ancestral diversification from common ancestors.

DOI: https://doi.org/10.7554/eLife.38278.003

underline that in CCB many different lineages of bacteria along with Cyanobacteria (*López-Lozano et al., 2012*) have an important role in the acquisition of nitrogen in the valley, contributing to the unbalanced nutrient stoichiometry. The diversity within this small scale in the Churince is immense, in particular when compared to 343 other studies based also on 16S rRNA microbiomes. Those studies comprised several contrasting environments: microbes associated to human, plant, soil, sediments, biofilm, marine biofilms, and extreme environments like Yellowstone hot springs, Guerrero Negro salt flats, and Antarctica soils, for all of which data is available in public databases (*Supplementary file 1*).

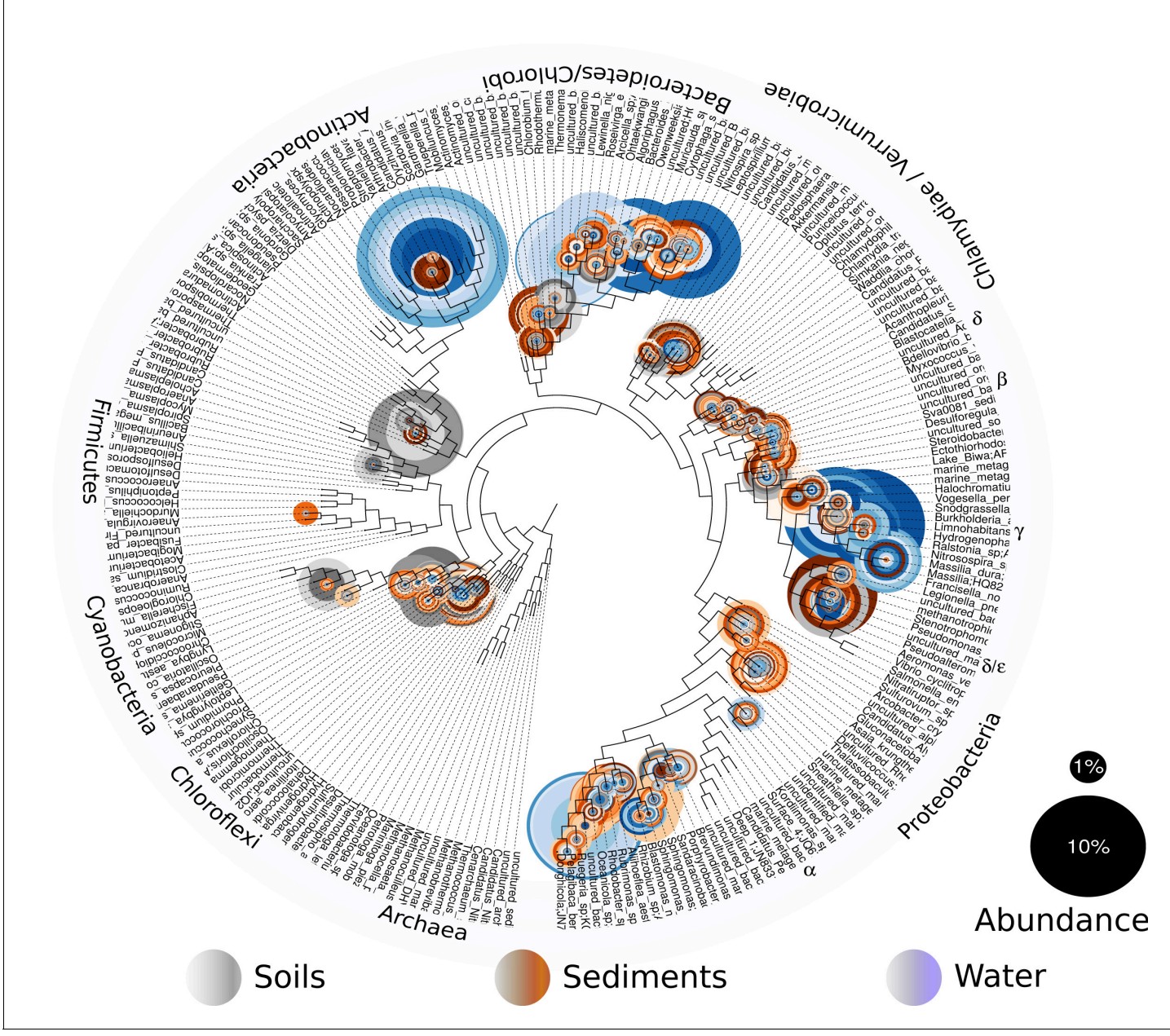

**Figure 2.** Overall prokaryotic diversity in Churince. Major phyla abundances in CCB is depicted in bar plots, only the most abundant phyla are shown, but there are 60 phyla present in CCB which are roughly 66.28% of known prokaryotic phyla, in a single location few meters away. Proteobacteria is the most abundant phylum, followed by Bacteroidetes, and Actinobacteria. Some phyla like Planctomycetes, Cyanobacteria, Acidobacteria, Chlorobi, and Firmicutes are more abundant in the sediment and soil-associated samples than in water columns. Each CCB sample is colour-coded according to its origin: blue for water; brown for sediments; and grey for soils.

DOI: https://doi.org/10.7554/eLife.38278.004

To further compare with the other environments, we used the individual OTUs and then computed their alpha diversity using Shannon index and Simpson function. Alpha diversity measures the number of species and their proportion within each of the sampling sites. Shannon index calculates diversity and abundance, though it is a poor predictor of diversity when rare species constitute a substantial part of the diversity (which is our case). Simpson's function calculates diversity based on the total number of species but does not take into account their relative abundances. The most diverse environments, according to Shannon's diversity index (Table 1S), are the aquatic sediments

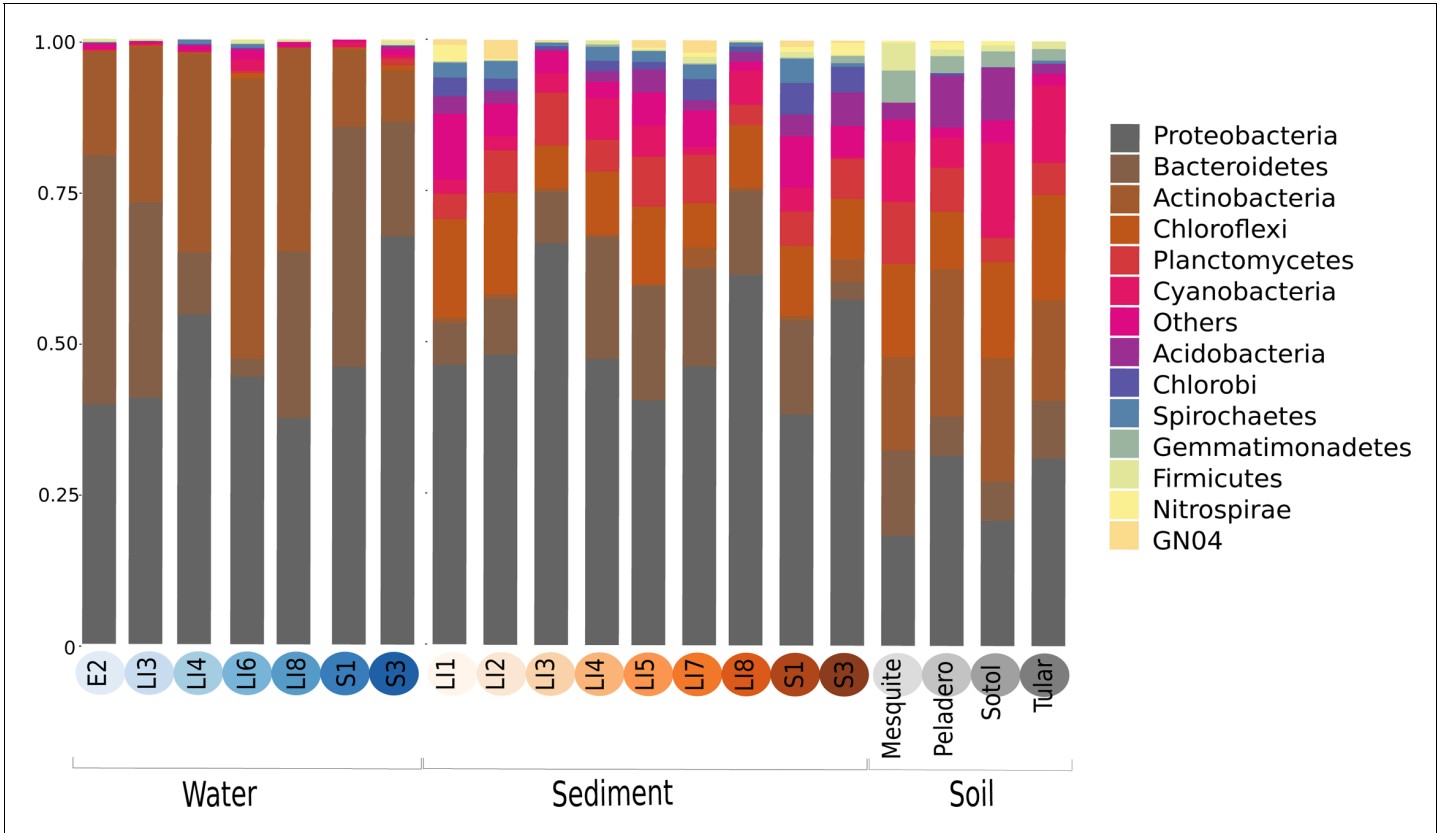

**Figure 3.** The phylogenetic placement per sampling site shows the names and abundances of the most represented genera within CCB samples. Even though each site has a particular profile, we can see the aggrupation by type of sample: soil, sediment and water.

DOI: https://doi.org/10.7554/eLife.38278.005

of different sites in the world; accordingly, in our dataset the Churince's sediment had the highest Shannon value. This was also supported by the Simpson's index that showed, as expected, that sediment was more diverse than water (Table 1S). Even though most of the diversity at Churince had low abundance (*Figure 2*, see the size of the circles in the phylogenetic tree), both Shannon's and Simpson's diversity indices revealed a very high microbial diversity, even when compared with other microbial diversity hotspots, such as Pearl river in China, or Guerrero Negro in Mexico. We suggest that the explanation for such a large diversity within such a small place is, in part, niche stability over geological times, and in addition, the diversification process reinforced by local adaptation.

To test for the lost world clade diversification, we focused on the diversity of bacteria from a single well-known genus, *Bacillus,* that are easily cultured. From our collection of approximately 2500 cultured *Bacillus* spp. from CCB, 16S gene sequences were obtained and compared to sequences in databases. We obtained 265 unique sequences selected at 97% identity, a very conservative estimate for *Bacillus*. In a global tree (*Figure 4*) with 1019 other OTUs reported for *Bacillus* spp. from around the world, we can observe the overall distribution and genetic distance of these CCB 16S sequences in relation to all known *Bacillus* lineages. We noticed that CCB strains formed multiple endemic (only found in CCB) lineages most of them with very deep branches, and that our sample increased by nearly 21% the number of previously known *Bacillus*.

Within the *Bacillus* spp. from CCB, we can distinguish two diverse sets of endemic lineages: one from sediment and another one closely related to marine *Bacillus* spp. (*Figure 4*). CCB *Bacillus* spp. from sediments are significantly older than the marine related CCB lineages, and according to our analyses calibrated using the divergence between the genus *Bacillus* and *Geobacillus* (*Moreno-Letelier et al., 2012*; *Battistuzzi and Hedges, 2009*) (*Figure 5*) may date back to the Ediacaran (635–541 mya, at the end of the Precambrian). The Ediacaran period, marks the start of the oxygenation of the ocean allowing not only the first animals to evolve (*Planavsky et al., 2010*), but also the

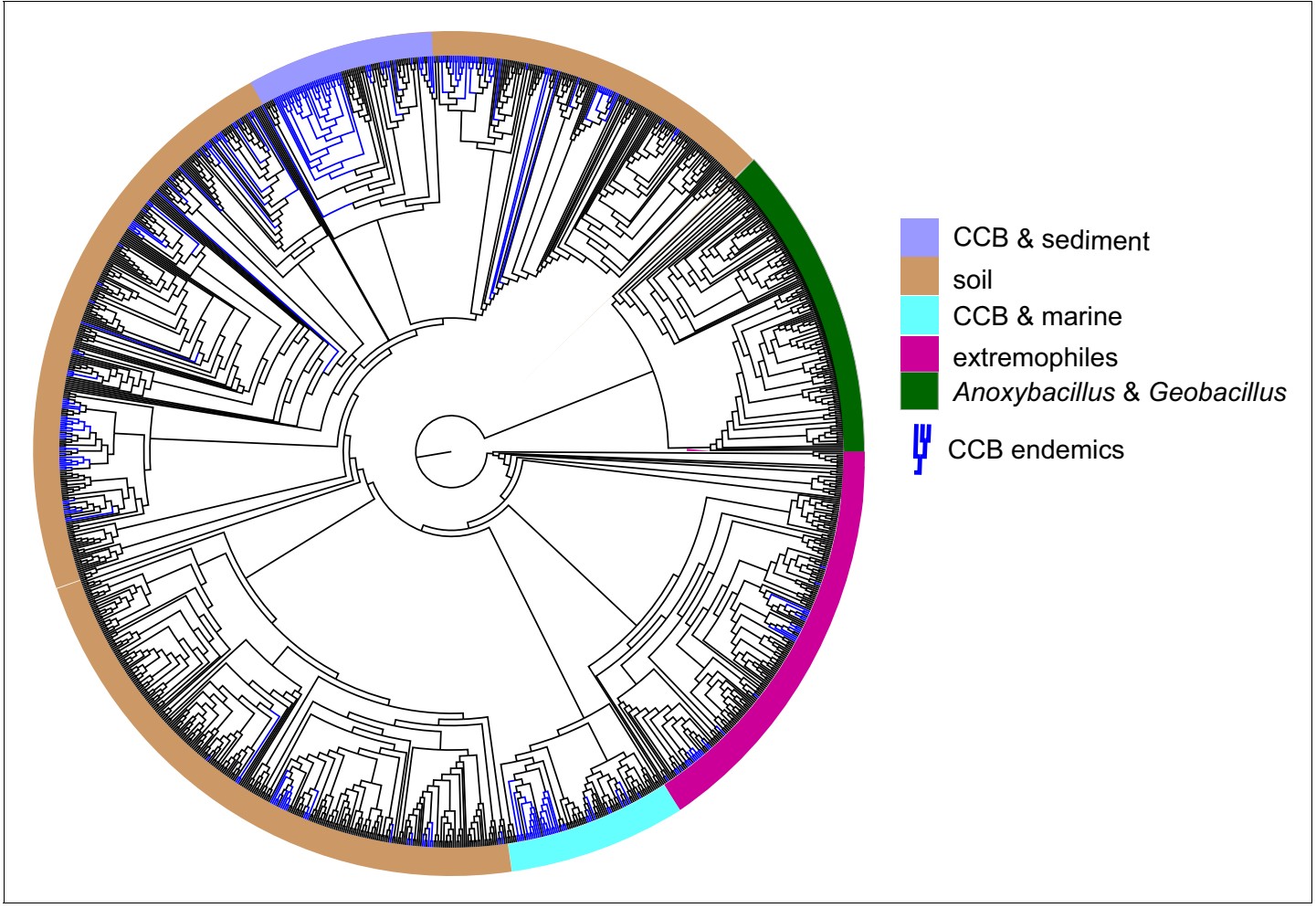

**Figure 4.** Dendrogram of the 1284 strains of the genus *Bacillus* reconstructed from 16 s rRNA. All taxa have a sequence divergence over 97%. Strains found in Cuatro Cienegas Basin (CCB) are denoted in blue. All other strains, including the outgroups, are denoted in light gray. The position of the sediment CCB *Bacillus* and marine CCB *Bacillus* lineages within the genus *Bacillus* are also indicated. The outgroup includes strains of *Geobacillus* and *Anoxybacillus..*

DOI: https://doi.org/10.7554/eLife.38278.006

The following figure supplement is available for figure 4:

**Figure supplement 1.** Bayesian phylogenetic trees of the order Clostridiales and the phylum Bacteroidetes reconstructed with 16 s rRNA.
DOI: https://doi.org/10.7554/eLife.38278.007

first aerobic Bacilli to diversify. Unlike the sediment lineage, CCB *Bacillus* species from water, did not form a monophyletic group, which suggests independent synchronized origins dating to the late Jurassic (*Figure 6*). A mixed representation of many lineages appears to have entered the CCB 'multidimensional niche bubble' simultaneously and did not become extinct. The presence of endemic and early divergent clades, with deep branches, of sediment and marine related CCB lineages, as well as the low extinction rates observed in these lineages (*Figure 7*), provides strong evidence for the lost world scenario (*Figure 1D*).

Why do we observe only two pulses of marine migration at CCB? We think that the first pulse can be explained by the abrupt change of nutrient stoichiometry balance at the end of the Ediacaran (*Elser et al., 2006*; *Planavsky et al., 2010*; *Stüeken and Kipp, 2017*). However, the second pulse is more puzzling since it implies the migration of many independent lineages into the CCB shore. It is possible that the reason is tectonics, as CCB is the point where the birth of the Tethys sea occurred in the western point of Pangea breakage (*Wolaver et al., 2013*; *Souza et al., 2006*).

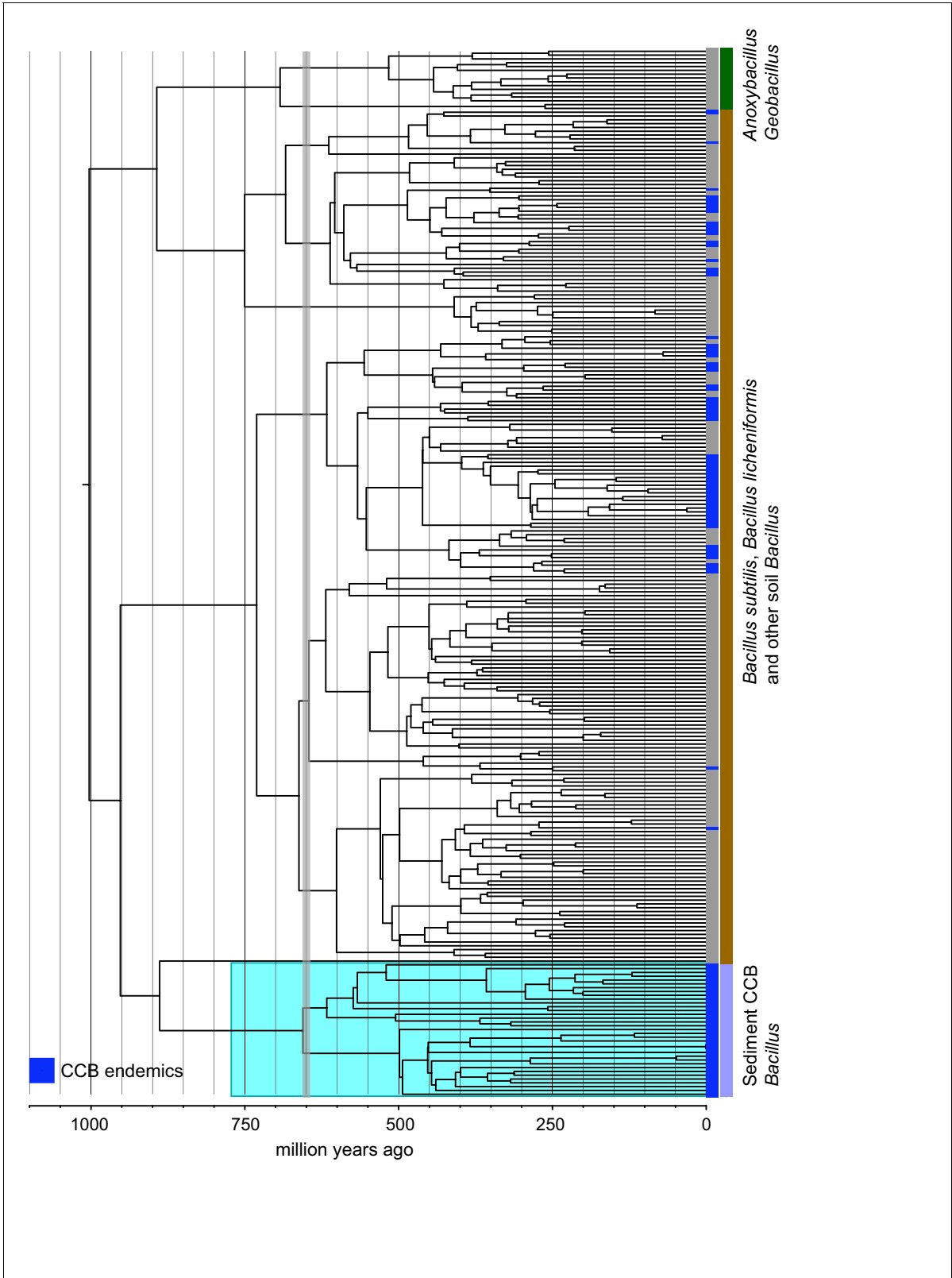

**Figure 5.** Dated Bayesian phylogeny of soil and sediment *Bacillus*, including the endemic lineage of sediment CCB *Bacillus* (highlighted in cyan). Strains endemic to CCB are denoted in blue. The vertical grey line indicated the date of divergence of sediment CCB *Bacillus* approximately 655 Ma, in the late Precambrian, during the Cryogenian period.

DOI: https://doi.org/10.7554/eLife.38278.008

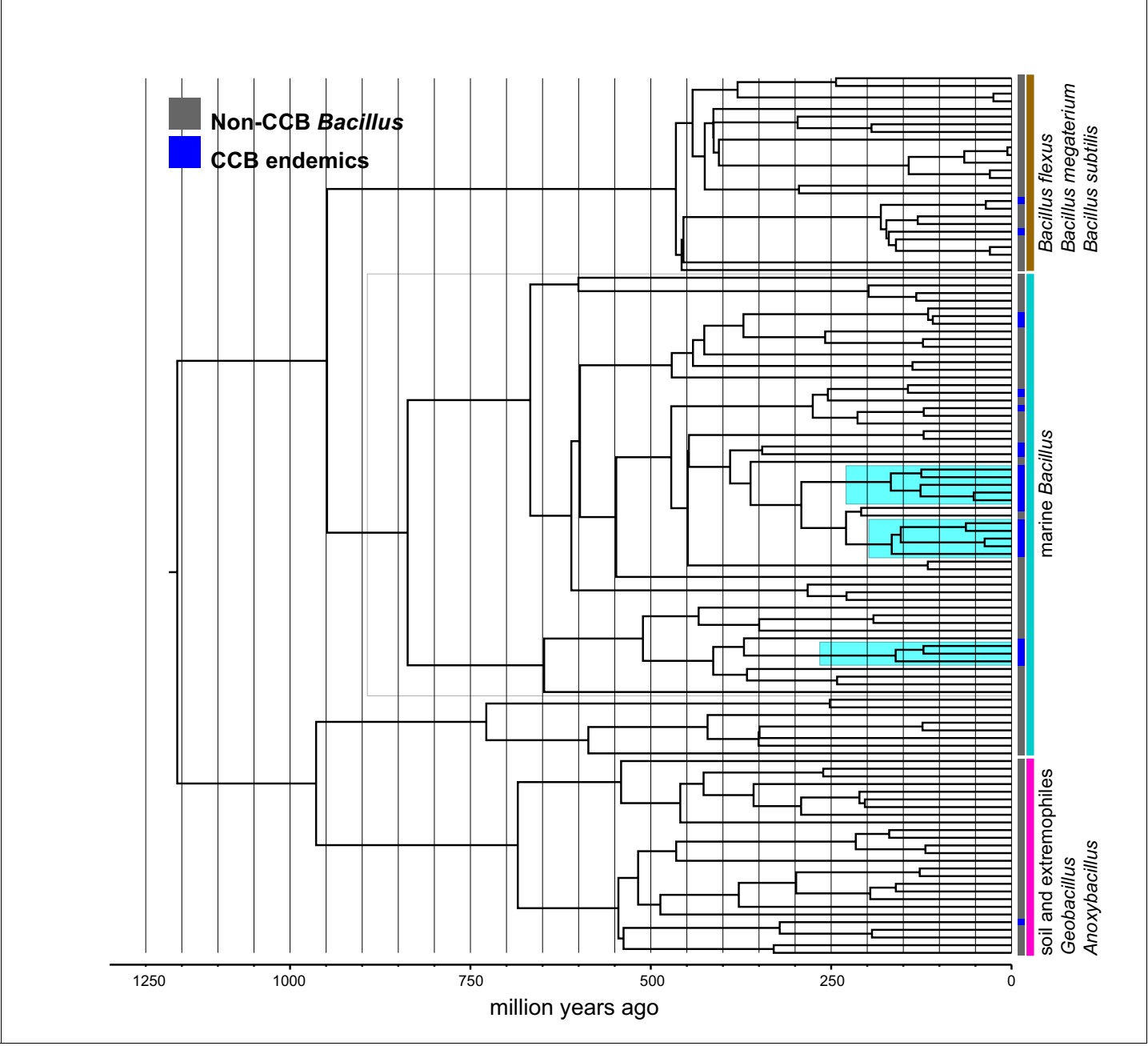

**Figure 6.** Dated Bayesian phylogeny of marine *Bacillus*, including endemic lineages from CCB (highlighted in cyan). The grey line indicates the divergence time of three independent CCB marine strains at around 160 Ma in the Late Jurassic period.

DOI: https://doi.org/10.7554/eLife.38278.009

Even though none of the other lineages of *Bacillus* endemic to CCB have been reported at any other site, they seem to constitute punctuated events of arrival to the 'island-like' niche and to have subsequently diversified locally. In contrast, in a study of *Bacillus* spp. from diverse environments in India many cosmopolitan *Bacillus* spp. were collected that had short branches to sister species (*Yadav et al., 2015*). When we compared the *Bacillus* sample with a much smaller sample of anaerobic Clostridiales and Bacteroidetes strains, the later were fewer and did not form monophyletic groups endemic to CCB, even though some of their branches were early divergent (*Figure 4—figure supplement 1*).

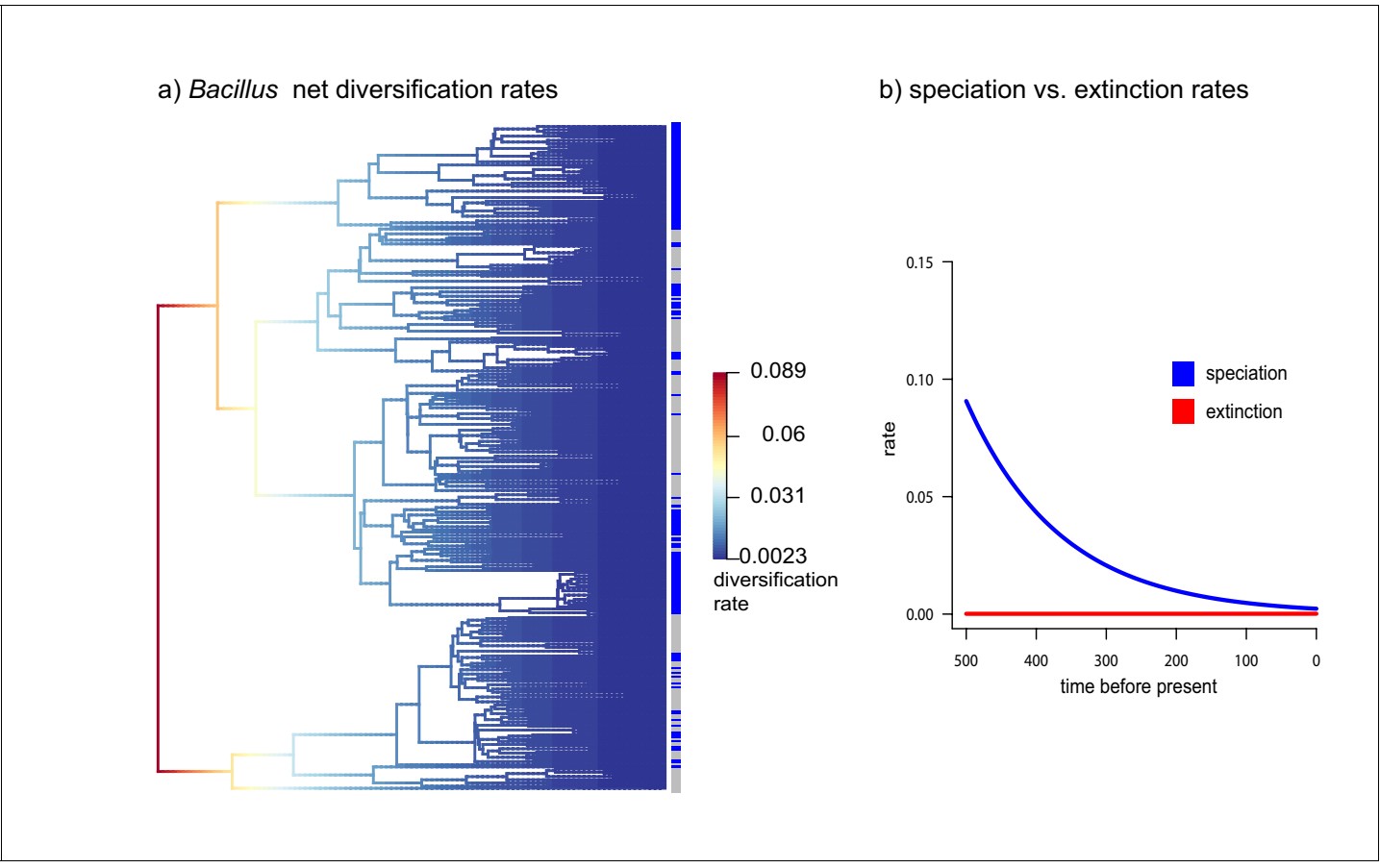

**Figure 7.** Diversification vs Extinction rates in *Bacillus* from CCB. (**a**) Net diversification rates in soil and sediment *Bacillus* plotted in a phylogenetic tree. (**b**) Speciation and extinction rates of soil and sediment Bacillus. Speciation rates lower over time, while extinction rates remain constant.

DOI: https://doi.org/10.7554/eLife.38278.010

## Discussion

Environmental conditions are known barriers to dispersal. The extreme unbalanced nutrient stoichiometries at CCB could certainly constrain immigration from phosphorus demanding populations that require a proportion of nitrogen to phosphorus closer to 16:1 (*Elser et al., 2006*) for their survival. This is the case of Patagonia's isolated and oligotrophic lakes with unique microbial communities with low diversity (*Aguayo et al., 2017*). CCB microbial populations are not only unique, they are also very diverse, despite CCB having an extremely unbalanced nutrient stoichiometry. Our data showed that the small and endangered Churince system in CCB contains 57 out of the 86 known Bacteria phyla, which is 66.3% of the recorded bacterial diversity at phylum level (data from 342 analysed microbiomes, Table 2S). This diversity is only comparable to Pearl River in China (*Wang et al., 2012*), where 48 microbial phyla were found. The Pearl River is a highly productive environment that receives inputs from multiple sources in the 2,400 km extension of China's third largest river. In contrast, Churince is a small hydrological system fed by a spring and extends 1 km at most.

The local scale makes the bacterial diversity of CCB even more interesting. Red Queen evolutionary processes have been shown to cause highly localized adaptation in numerous systems, including microbial ones. The Red Queen process proposes that organisms continuously adapt to changing conditions, particularly those involving antagonistic interactions within and between species, causing an increase in localized adaptation and coevolution (*Lawrence et al., 2012*). Experiments of competition between strains of *Bacillus* from different sites within Churince showed marked antagonisms against different strains even from sites a few meters away (*Pérez-Gutiérrez et al., 2013*), consistent with the Red Queen model. We have also observed extreme antagonism in the case of CCB

Actinobacteria with non-CCB bacteria (*Arocha-Garza et al., 2017*), which may prevent migration from bacteria outside the basin.

What circumstances can keep the marine signal intact? This requires prevention of both genes and populations from migrating (*Souza et al., 2008*; *Souza et al., 2012*). Indeed, we have observed that most of the bacterial lineages are local and clonal (*Cerritos et al., 2011*; *Avitia et al., 2014*). Genomes are small and have few imprints of horizontal gene transfer (HGT) (*Alcaraz et al., 2008*; *Alcaraz et al., 2010*; *Gómez-Lunar et al., 2016*). The only exception is *Vibrio* from CCB, a lineage that thrives in perturbed, higher nutrient environments within the basin, and that display recombination rates similar to the ones observed in marine *Vibrio*. The caveats being, that such recombination is mostly homologous, maintaining local adaptation and genetic isolation (*Gómez-Lunar et al., 2018*). Even though most CCB lineages are clonal, the rare HGT events allow linkage disequilibrium to break. Hence, we do not see small populations sizes or genetic sweeps as expected in completely clonal lineages (*Cohan, 2016*). Moreover, extinction rates are low, while diversification rates are high in some lineages at certain times (*Figure 7*).

We believe that selective sweeps have not purged the genetic diversity that would drive lineages to extinction, and involves Black Queen dynamics, in addition to the Red Queen processes discussed above. Tolerance and cooperation between strains are observed. For instance, strains of endemic lineages of *Bacillus* from Churince require cross feeding and cooperation to obtain even amino-acids (*Rodríguez-Torres et al., 2017*). These cross-feeding observations fit a 'Black Queen' model (*Morris et al., 2012*) where adaptation to severely limited resources lead to genomic streamlining and metabolic co-dependency (*Alcaraz et al., 2008*; *Gómez-Lunar et al., 2016*). Hence, our potential explanation for the long-term survival of lost world *Bacillus* species in CCB involves multiple eco-evolutionary feedbacks. Migrants are suppressed by antagonistic coevolution and community cohesion that is maintained by co-dependent metabolic interactions. Moreover, although *Bacillus* spp. can form spores, the ultimate strategy to survive stressful conditions, we have shown that CCB *Bacillus* spp. are competing actively in the microbial communities (*Pérez-Gutiérrez et al., 2013*). Antagonisms and cooperation occur simultaneously in microbial mats and stromatolites (*Anda et al., 2018*), and it is possible that the same dynamics occurred originally in the South-Western shores of Laurentia (*Kershaw, 2017*).

All our results suggest that extreme unbalanced nutrient stoichiometry, along with community cohesion function like a 'semipermeable' barrier to migration, where effective migration is possible, but rare. Fossil evidence shows that stromatolites were still abundant between the Permian and Triassic boundary, in the site where the Tethys sea opened in the South-Western shores of Laurentia (*Kershaw, 2017*) where CCB was located at the onset of the Mesozoic (*Wolaver et al., 2013*; *Souza et al., 2006*). However, during the massive extinction event that marked the end of the Permian, stromatolites became rare, except on the western shores of the Tethys sea (*Kershaw, 2017*). Microbial mats and stromatolites can still be found in other sites of the planet, but at CCB, aside from giving testimony to the past, microbial lineages have been safeguarded, bringing evidence for a lost world. The extreme stoichiometric imbalance in the Churince can be explained in part due to the very old weathered rocks that have lost their phosphorous and, from the entrance of nitrogen into the system mainly through nitrogen fixation by many members of the community (*Lee et al., 2017*; *López-Lozano et al., 2012*).

Even though CCB microbial communities have survived for an extended period of time, their particular niche conditions are being destroyed in the Anthropocene. This impact is even more poignant because CCB wetland has shrunk 90% in the last 50 years, and its deep aquifer has been devastated by the use of fossil water in local agricultural practices. This deep niche change has already destroyed many of the microbial complex communities in Churince. However, we believe that this change can be reversed if the channels that drain the wetland are closed and the wetland recovers its water cycle.

The CCB has a population of 14,000 people, as well as flourishing tourism, which represent an input of large amounts of nitrogen and phosphorus. Fortunately, most of the human activities and nutrient inputs occur at least 20 km away from the oasis and their astounding microbial communities. We believe that calcium carbonate rocks have worked as buffer between the human activities and the turquoise blue ponds (*Wolaver et al., 2013*). Nevertheless, our experimental evidence shows that an increment in nutrients results in algal blooms and a reduction or disappearance of endemic lineages (*Lee et al., 2017*). Hence, if such mineral buffer gets saturated or the wetland disappears,

the environmental singularity that makes CCB unique can change, erasing the biological evidence of this 'lost world'.

We hope that awareness of this problem will push for proper measures for a change in agricultural practices and sewage management by the county and State authorities. Conservation of the unique niche in CCB and similar sites is paramount for our understanding of the deep past as well as to predict and protect the future of our planet.

## Materials and methods

### Microbial diversity context

We sampled ten sites during May 2011 in the Churince system of CCB in a 300 m long lagoon plus two more sites in the spring-head, ca. 1 km away (latitude: 26° 50' 53.19' N, longitude: 102° 8' 29.98' W). For each site, permission to sample was obtained from the federal government in Mexico (SEMARNAT, dirección de vida silvestre FAUT0230). In each sample site, we took 50 g of sediment and a gallon of water as well as a sample of both for biogeochemical variables, nutrients and minerals: C, N, P, Ca, Mg. We also sampled four types of vegetation from an established gradient and obtained composite soil samples. We extracted DNA from each sample using the same methodology (*López-Lozano et al., 2012*). Metagenomic DNAs were sent to JCVI (San Diego, CA, USA) for 16S rRNA amplicon gene library (341 F-926R primers) 454 pyrosequencing (Roche, Brandford, Ct, USA).

A total of 950,000 reads were sequenced; we required a minimum of 50,000 reads per site, with a minimum 500 bp length after Quality Control check. Not all samples produced the same amount of sequences, probably due to the natural low yield of DNA extraction in CCB water and sediments. Nevertheless, even at 97%, diversity is high, encompassing all the know phyla of Bacteria but a very low diversity and abundance of Archaea and mostly none of the cosmopolitan human related microbial taxa.

The 16S rRNA gene analysis was done as previously reported (*Avitia et al., 2014*; *Alcaraz et al., 2016*). Briefly sequencing quality was processed and filtered using FASTQ and Fastx-toolkit, we filtered out any sequence with Phred < 30, length <500 bp. Operational taxonomic units (OTUs) were clustered using cd-hit-est (*Huang et al., 2010*) with a 97% identity threshold cut-off. The OTUs were parsed into QIIME pipeline and the taxonomic assignments were done against Greengenes DB (v 13.8 (*DeSantis et al., 2006*)). Chimeras were removed after taxonomic assignments and detected by ChimeraSlayer (*Human Microbiome Consortium et al., 2011*). Data management, diversity statistic, and plots were done using R phyloseq package (*McMurdie and Holmes, 2014*) and ggplot2 and RColorBreweer R libraries. Pplacer (RRID:SCR_004737) was used to place the diversity into a reference tree (*Figure 2*) (*Matsen et al., 2010*). We are using diversity indexes, rather than OTU comparisons because of differences in sequencing technologies, primers used for 16S rRNA gene, coverage depth, and other factors that could affect an overall OTU comparison among different studies.

Compared datasets (342) were retrieved from public available databases like NCBI's SRA, MG-RAST, and HMP (Human microbiome project) websites. Detailed information about accessions used is available as supplementary material in Table S1.

### *Bacillus* tree

The sequence identity clustering of all 16S rRNA gene sequences from the genus *Bacillus* spp. and sister genera *Anoxybacillus* and *Geobacillus* were retrieved from online databases Ribosomal Database Project (RRID:SCR_006633) and Genbank (RRID:SCR_002760), 1019 of them at 97% sequence identity, plus 648 sequences of cultivated *Bacillus* spp. from CCB (accession numbers in *Supplementary file 2*) selected with the same criterion out of more than 2500 cultivates strains; sequence clustering was done with cd-Hit (http://weizhongli-lab.org/cd-hit/, *Huang et al., 2010*, RRID:SCR_007105).

These sequences were further aligned with the 16 s rRNA sequences from CCB with the MUSCLE (RRID:SCR_011812) plugin in Geneious (*Kearse et al., 2012*)(RRID:SCR_010519). Neighbour-joining trees were constructed using genetic distances, with the ape and seqinr R package. The sequences were also aligned with the 16 s rRNA sequences from CCB with the MUSCLE plugin in Geneious 5.4.6 (*Kearse et al., 2012*). In order to have a control, a subset of OTUs from Clostridiales (n = 131;

18 unique from CCB) and Bacteroidetes (n = 189; 12 unique from CCB, Genebank numbers in Table S2) was then used to construct Bayesian phylogenies including CCB cultivated anaerobic strains while for *Bacillus*, we only focused on a mayor clade which had a better representation from CCB samples, henceforth called sediment and soil *Bacillus* (n = 311) and another with predominantly marine *Bacillus* (n = 115; Genebank numbers in Table S2).

Phylogenies were reconstructed using BEAST v. 1.8.2 (*Drummond et al., 2012*) (RRID:SCR_011812), with a Birth-Death speciation model, relaxed lognormal clock models and the following substitution models Bacteroidetes (HKY + I + G), Clostridiales (HKY + I + G), and both *Bacillus* clades (GTR + I + G). All substitution models were chosen using a Bayesian Information Criterion on likelihoods estimated with jModeltest 2.1.7 (*Darriba et al., 2012*)(RRID:SCR_015244). Three separate runs were performed for each dataset of 50 million chains each and then combined using Log-Combiner v1.8.2. Parameter convergence was evaluated using Tracer v. 1.6.0. Ultrametric trees were obtained with relative node ages, which were later scaled to produce ultrametric trees with absolute ages to be used in the diversification rate analyses using the R package phytools (*Revell, 2012*).

The calibration points to date all trees were obtained from literature. The calibration point of Clostridiales was set at 3,500 Ma (*Battistuzzi and Hedges, 2009*) with a normal distribution at the root of the tree, for Bacteroidetes was set to 2,500 Ma (*Battistuzzi and Hedges, 2009*), also with a normal distribution. The node heights of the sediment *Bacillus* lineage were obtained by estimating the divergence dates within the genus *Bacillus* using a smaller phylogenetic sampling. The analysis was conducted using BEAST v. 1.8.2 (*Drummond et al., 2012*) and the calibration points were set at the divergence of the genus *Bacillus* from *Geobacillus* at a conservative 1,144.7 Ma (sd = 164) (*Moreno-Letelier et al., 2012*) following the great oxidation event, set to a normal distribution. Another calibration point was set in the diversification of *Bacillus* at 1047 Ma (sd = 159), also set to a normal distribution, as it is the recommendation when using node ages estimated by molecular dating (see BEAST v.1.8.2 documentation). The clock model was a log normal relaxed clock and the analysis was run in BEAST v. 1.8.2 (*Drummond et al., 2012*).

Changes in diversification rates were estimated with a Bayesian framework using BAMM 2.5.0 (*Rabosky, 2014*). This method estimates the speciation rates, identifies shifts along the phylogeny and estimates the confidence intervals of the various shift configurations detected using a Markov Chain Monte Carlo to explore the universe of candidate models (*Rabosky, 2014*). The analyses were carried out using the scaled ultrametric trees of sediment *Bacillus*, Bacteroidetes and Clostridiales, with 10 million generations for all cases except Clostridiales, which required 20 million generations to reach convergence and adequate effective sampling sizes. Priors were estimated using the function setBAMMpriors implemented by the R package BAMMtools. Results were analysed using BAMMtools on R (*R Development Core Team, 2014*; RRID:SCR_001905) to obtain the best shift configuration, Bayes factors of number of shifts and posterior probabilities of shifts distributions.

Finally, we compared the distribution of rates along the tree of all lineages in order to assess the relative diversification rate differences in all lineages.

## Acknowledgments

This research was supported by funding from WWF-Alianza Carlos Slim, Sep-Ciencia Básica Conacyt grant 238245 to both VS and LEE, and Sep-Ciencia Básica Conacyt grant 220536 to GOA. The paper was written during a sabbatical leave of LEE and VSS in the University of Minnesota in Peter Tiffin and Michael Travisano laboratories, respectively, with support of the program PASPA- DGAPA, UNAM. We thank Laura Espinosa-Asuar for making *Figure 1* and Chris Dupont of JCVI for the phylogenetic analysis in *Figure 2*. We thank Africa Islas for excellent technical assistance.

## Additional information

### Funding

| Funder | Grant reference number | Author |
|---|---|---|
| WWF International | desunam7 | Valeria Souza |

| Consejo Nacional de Ciencia y Tecnología | 238245 | Valeria Souza |
| Consejo Nacional de Ciencia y Tecnología | 220536 | Gabriela Olmedo |

Fundacion Carlos Slim along with WWF Mexico funded a 5 year study of the biodiversity of Cuatro Cienegas in order to obtain the total inventarium of this extremely important and endangered site. They did not dictate any rules on the research or hold any rights on its publication.

## Author contributions

Valeria Souza, Conceptualization, Resources, Supervision, Funding acquisition, Investigation, Writing—original draft, Project administration, Writing—review and editing; Alejandra Moreno-Letelier, Data curation, Software, Formal analysis, Investigation, Methodology, Writing—review and editing; Michael Travisano, Conceptualization, Investigation, Visualization, Writing—original draft; Luis David Alcaraz, Data curation, Formal analysis, Methodology, Writing—review and editing; Gabriela Olmedo, Conceptualization, Data curation, Funding acquisition, Investigation, Methodology, Writing—original draft, Writing—review and editing; Luis Enrique Eguiarte, Conceptualization, Supervision, Funding acquisition, Investigation, Writing—review and editing

## Author ORCIDs

Valeria Souza https://orcid.org/0000-0002-2992-4229
Alejandra Moreno-Letelier https://orcid.org/0000-0001-7524-7639
Michael Travisano http://orcid.org/0000-0001-8168-0842
Luis David Alcaraz http://orcid.org/0000-0003-3284-0605

## Decision letter and Author response

Decision letter https://doi.org/10.7554/eLife.38278.015
Author response https://doi.org/10.7554/eLife.38278.016

## Additional files

### Supplementary files

• Supplementary file 1. Churince's diversity by site.
DOI: https://doi.org/10.7554/eLife.38278.011

• Supplementary file 2. Genebank accessions numbers of strains used in this study.
DOI: https://doi.org/10.7554/eLife.38278.012

• Transparent reporting form
DOI: https://doi.org/10.7554/eLife.38278.013

### Data availability

Bacteroidetes and Clostridiales 16S data is available in GenBank (Accession numbers in Supplementary File 2). Additional Bacillus sequences were obtained from RDP (https://rdp.cme.msu.edu/). All CCB endemic Bacillus, Bacteroidetes and Clostridiales strains were submitted to GenBank (accession numbers in Supplementary File 2). Metagenomic 16s data from Churince is deposited at MG Rast (https://www.mg-rast.org/), accession numbers in the Supplementary File 2.

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
