## [Decision Letter]

Thank you for submitting your article "The lost world of Cuatro Cienegas Basin, a relictual bacterial niche in a desert oasis" for consideration by *eLife*. Your article has been reviewed by three peer reviewers, including Roberto Kolter as the Reviewing Editor and Reviewer #1, and the evaluation has been overseen by Ian Baldwin as the Senior Editor. The following individual involved as another reviewer of your submission has agreed to reveal his identity: Fred Cohan (Reviewer #2).

The reviewers have discussed the reviews with one another and the Reviewing Editor has drafted this decision to help you prepare a revised submission.

Summary:

This paper provides support for the concept that some habitats (in this case Cuarto Ciénagas in Mexico), due to their particular physicochemical properties have caused microbial communities to remain in virtual isolation for millions of years. This has led to an independent evolution of high diversity bringing forth the idea of a biological "lost world." A major strength of this work is the ability to connect branching events with geologic time.

The authors describe a relict bacterial community, in which some endemic lineages appear to have been diversifying in situ for hundreds of million years. The paper also makes a reasonable point (although we think it could be made better) that migration is hindered more by failure of invading bacteria to thrive rather than failure to disperse. Also, the paper lays out an original protocol for testing the lost world hypothesis, although we believe that this point needs to be made clearer. That 1/4 of the diversity in the *Bacillus* genus is found in this study system is amazing. Overall the hypothesis is very intriguing, and the supporting evidence is encouraging.

This is an important scientific study that, in addition, should prove very useful in policy-making in terms of possible management of this and other endangered ecosystems. We are thus very positive about this manuscript. However, there are some significant revisions that will be needed to make this manuscript ready for publication in *eLife*.

Essential revisions:

1) The authors should lay out their rationale for testing the lost world hypothesis more clearly. What is not stated clearly in the Introduction is that the Galapagos model is distinguished experimentally from the lost world model by the depth of endemic lineages. They should state this explicitly.

2) In general, the figures are not as helpful as they should be. The one figure that most clearly drives home the major point is Supplementary Figure 3. This suggests that one endemic *Bacillus* clade goes back 655 million years! Also, Supplementary Figure 4 is especially interesting, showing that there are three endemic lineages that are around 160 million years old. Why are these in the supplement? These are the clearest and most compelling figures in the paper. We strongly suggest that the authors pay attention to the fact that *eLife* does not encourage supplementary figures. In revising the manuscript, the authors should revise their figures such that the information in current Supplementary Figures 3 and 4 is emphasized since they provide the strongest evidence for the lost world hypothesis. In contrast, current Figures 2 and 3 describe overall diversity of the whole community or the *Bacillus* genus and are less important for examining the lost word hypothesis. In addition, Figure 2 is confusing, we don't understand what the rings of blue color are. The caption should state the taxon level analyzed in the diversity indexes.

3) The age of the longstanding endemic *Bacillus* lineage is really important. The authors should make this point more emphatically by briefly mentioning the calibration point for their time estimate.

4) Parts B and C of Supplementary Figure 2 seem irrelevant. Very few of the lineages shown in the Bacteroidetes and Clostridiales are endemic, and so the rates of diversification shown here are largely determined outside the CCB system. So, the authors' focus on how the rates of diversification in these taxa have changed over time does not seem relevant to the CCB story.

5) The authors should cite experimental data that indicate that migration is hindered by failure of invading strains. Or if the data do not exist, the authors should suggest some relevant experiments, briefly.

6) The authors should state explicitly how the Black Queen model would apply to the CCB system.

7) The authors should report the salinity of CCB, to help explain why marine organisms would do well here. And, in general, it would be very useful to more extensively describe the physico-chemical properties of CCB that make it such a unique environment (see also our last point on phosphorus concentration).

8) In the Abstract, the lost world is described as having low extinction rates. This needs to be further developed in the manuscript. Our take on the evidence supporting this claim is that the endemic CCB *Bacillus* lineages have not been found anywhere else on modern earth, and are 100 million years old or older, which is much older than the time that the CCB shifted from a coastal to an inland location, 35 million years ago. Presumably these lineages were present in the rest of the world 35 million years ago, but went extinct everywhere else but this isolated lost world? What might have triggered their extinction elsewhere that was not present at CCB?

9) In the fourth paragraph of the main text, the authors mention that the lost world is expected to have lineages that are "separated" from other lineages. Since the study is tracking one phylogenetic marker gene (16S), which ideally does not participate in horizontal gene transfer, this is not absolute indication that genetic isolation is occurring. In contrast genomic/metagenomic analyses could help determine if these CCB endemics are not sharing genes with the rest of the world. If these analyses are available, the authors should include. If not, the authors should at least discuss this possible approach in the concluding paragraphs of the manuscript.

10) Finally, we are very concerned that a 0.5 mM phosphorus concentration is being put forth as "extreme oligotrophy." We are hoping this is just a typo on the part of the authors because 0.5 mM phosphorus is ~500,000X higher than the concentration reported in the Atlantic Ocean, where P has been empirically shown to limit microbes, and 50,000X higher than in the Pacific, where P is not the limiting nutrient (it's usually N or Fe) (see Karl 2002 Trends Micro 10: 410 as one review). The authors should elaborate more on how the actual phosphorus concentration in Cuatro Ciénagas compares to other habitats rather than simply state it is low (although if it really is 0.5 mM, then it is by no means low!). A lot of the paper rests on the uniqueness of the phosphorus concentration. This fact needs to be reassessed and/or elaborated further.

[Editors' note: further revisions were requested prior to acceptance, as described below.]

Thank you for submitting your article "The lost world of Cuatro Cienegas Basin, a relictual bacterial niche in a desert oasis" for consideration by *eLife*. The Reviewing Editor has read the revised version and decided against sending it for re-review because it was felt this new version addressed the vast majority of the concerns raised after our initial review.

Summary:

This is a very important set of results and the Editor applauds the extensive revisions already performed. Unfortunately, the manuscript still needs a fair amount of text corrections before acceptance.

Essential revisions:

The main issue is that you refer to "extreme unbalanced stoichiometry" when, in my view, you really mean "extreme unbalanced in nutrient stoichiometry." Otherwise talking about stoichiometry is in a vacuum and makes no sense. If you prefer it could be universally "extreme unbalance in the phosphorus to nitrogen stoichiometry." But in either case, this correction should be made throughout and consistently.

For a paper this important, it was disappointing to see errors such as using "where" when it should have been "were." Please be sure to submit a version where you have gone over the text multiple times "with a fine toothed comb."

---

## [Author Response]

Essential revisions:1) The authors should lay out their rationale for testing the lost world hypothesis more clearly. What is not stated clearly in the Introduction is that the Galapagos model is distinguished experimentally from the lost world model by the depth of endemic lineages. They should state this explicitly.

In the actual version of the text we explain the model more carefully, changing the term “Galapagos” for “island model”, as this is more precise. This model is carefully explained now, both in the last paragraph of the Introduction and in the legend of Figure 1.

2) In general, the figures are not as helpful as they should be. The one figure that most clearly drives home the major point is Supplementary Figure 3. This suggests that one endemic Bacillus clade goes back 655 million years! Also, Supplementary Figure 4 is especially interesting, showing that there are three endemic lineages that are around 160 million years old. Why are these in the supplement? These are the clearest and most compelling figures in the paper.

We completely agree with reviewers, and moved the figures that were in supplementary materials to the main text. Now these former Supplementary Figures 3 and 4 are Figures 5 and 6.

We strongly suggest that the authors pay attention to the fact that eLife does not encourage supplementary figures. In revising the manuscript, the authors should revise their figures such that the information in current Supplementary Figures 3 and 4 is emphasized since they provide the strongest evidence for the lost world hypothesis.

Thank you for the comment. In the text now it is very clear that they are very important, see Results, fourth paragraph that later is retaken in the Discussion.

In contrast, current Figures 2 and 3 describe overall diversity of the whole community or the Bacillus genus and are less important for examining the lost word hypothesis. In addition, Figure 2 is confusing, we don't understand what the rings of blue color are. The caption should state the taxon level analyzed in the diversity indexes.

In order to have a clearer message for the audience, we spliced Figure 2 into new Figure 2 and Figure 3.

Figure 2’s improved legend reads: “Overall prokaryotic diversity in Churince. […] Bubbles in the tree are proportional to relative abundances and color-coded according to its origin: blue for water; brown/orange for sediments; and greys for soils”.

While Figure 3’s new legend indicates: “The phylogenetic placement per sampling site shows the names and abundances of the most represented genera within CCB samples. Even though each site has a particular profile, we can see the aggrupation by type of sample: soil, sediment and water.”

Diversity indexes are now in Supplementary file 1.

3) The age of the longstanding endemic Bacillus lineage is really important. The authors should make this point more emphatically by briefly mentioning the calibration point for their time estimate.

The lack of reliable fossil evidence for bacteria makes calibration complicated. Therefore, we used known geologic events to set calibration points for specific lineages.

In previous work cited in the text (Moreno-Letelier et al., 2012) we published a calibrated phylogeny of Firmicutes, calibrated using the Great Oxidation Event to mark the divergence between anaerobic and aerobic Firmicutes, we then estimated the divergence time between *Geobacillus-Anoxybacillus* and *Bacillus* and used that date (with its associated standard deviation) as a calibration point for all divergences within *Bacillus*. This is a method that is widely used in groups with scarce fossil record and has many limitations.

However, our main conclusions that the endemic lineages in CCB are very old still hold. In the new version, this is explained in Materials and methods, where calibration points are explained in the subsection “*Bacillus* tree”.

4) Parts B and C of Supplementary Figure 2 seem irrelevant. Very few of the lineages shown in the Bacteroidetes and Clostridiales are endemic, and so the rates of diversification shown here are largely determined outside the CCB system. So, the authors' focus on how the rates of diversification in these taxa have changed over time does not seem relevant to the CCB story.

We agree, and moved those two trees to the only remaining supplementary figure, Figure 4—figure supplement 1.

5) The authors should cite experimental data that indicate that migration is hindered by failure of invading strains. Or if the data do not exist, the authors should suggest some relevant experiments, briefly.

We agree, and included now several new citations that are relevant for the argument. See Discussion, second paragraph.

6) The authors should state explicitly how the Black Queen model would apply to the CCB system.

On one hand, we have a large part of the endemic lineages presenting auxotrophic metabolism, which implies a codependency with their community. On the other hand, we believe that cheaters are not allowed given the strong antibiotic response to all “foreigners”. Also, our hypothesis is that the very strong stoichiometric unbalance works as a filtering for most new comers. All these ideas in the improved text are in the fourth paragraph of the Discussion.

7) The authors should report the salinity of CCB, to help explain why marine organisms would do well here. And, in general, it would be very useful to more extensively describe the physico-chemical properties of CCB that make it such a unique environment (see also our last point on phosphorus concentration).

In the first paragraph of the Introduction, we indicate now that even if NaCl is relatively low, the concentration of other ions, such in particular of Mg^2+^, and SO_4_ are abundant enough to simulate ocean osmolality, this time citing several studies analyzing this in CCB.

As for the P concentration, yes, we are sorry for the typo, the concentrations were several orders of magnitude lower, in 0.5 μM not mM.

Now in order to make things clearer we explain in all the text that more than P by itself been low what is relevant is the stoichiometric unbalance in order to explain CCB lost world singularity.

8) In the Abstract the lost world is described as having low extinction rates. This needs to be further developed in the manuscript.

In this new version, Figure 7 is included in the main text. In it we explicitly show how extinction rates remained low and relatively constant. Our interpretation of the evidence supporting this claim is that the endemic CCB *Bacillus* lineages have not been found anywhere else on modern Earth, and are 100 million years old or older, which is much older than the time that the CCB shifted from a coastal to an inland location, 35 million years ago. Presumably these lineages were present in the rest of the world, but went extinct everywhere else except in this isolated lost world.

Interestingly, it seems that sediment *Bacillus* from CCB are part of the first radiation of aerobic *Bacilli*. This is explained now in the fourth paragraph of the Results and in the discussion of HGT in the third paragraph of the Discussion.

9) In the fourth paragraph of the main text, the authors mention that the lost world is expected to have lineages that are "separated" from other lineages. Since the study is tracking one phylogenetic marker gene (16S), which ideally does not participate in horizontal gene transfer, this is not absolute indication that genetic isolation is occurring. In contrast genomic/metagenomic analyses could help determine if these CCB endemics are not sharing genes with the rest of the world. If these analyses are available, the authors should include. If not, the authors should at least discuss this possible approach in the concluding paragraphs of the manuscript.

We agree and have now included this issue in the third paragraph of the Discussion where we have now several new citations of studies from microbes of CCB using other genes and additional analyses. in effect, we have many papers showing that clonality and local adaptation seems to be the rule and that similar patterns can be found using 16S and fingerprinting (Cerritos et al., 2010), MSLT approach (Rebollar et al., 2012; Avitia et al., 2015) as well as with complete genomes (Alcaraz et al., 2008; 2010; Goméz-Lunar et al., 2016 and Gómez-Lunar et al., 2018).

10) Finally, we are very concerned that a 0.5 mM phosphorus concentration is being put forth as "extreme oligotrophy." We are hoping this is just a typo on the part of the authors because 0.5 mM phosphorus is ~500,000X higher than the concentration reported in the Atlantic Ocean, where P has been empirically shown to limit microbes, and 50,000X higher than in the Pacific, where P is not the limiting nutrient (it's usually N or Fe) (see Karl 2002 Trends Micro 10: 410 as one review). The authors should elaborate more on how the actual phosphorus concentration in Cuatro Ciénagas compares to other habitats rather than simply state it is low (although if it really is 0.5 mM, then it is by no means low!). A lot of the paper rests on the uniqueness of the phosphorus concentration. This fact needs to be reassessed and/or elaborated further.

We agree completely and as we explained before, we had a mistake in the units of P concentration. We develop in this version the argument of unbalanced stoichiometry and very low proportion of P, since colimitation is common in poor ecosystem, the unbalance is unique, and it is evident even at the cellular level. This is included now this version all over the text.